

# Identification of demographic factors and health problems that affect the acceptance of disease and health behaviors of patients with osteoarthritis

Matylda Sierakowska[1], Izabela Wysocka-Skurska[2] and Wojciech Kułak[3]

[1] Department of Integrated Medical Care, Medical University of Bialystok, Bialystok, Poland
[2] Department of Rheumatology and Internal Diseases, University Hospital in Bialystok, Bialystok, Poland
[3] Clinic Rehabilitation Center for Children with Early Help Disabled Children "Give a Chance", Medical University of Bialystok, Bialystok, Poland

Corresponding author
Matylda Sierakowska, matyldasierakowska@gmail.com

## ABSTRACT

**Introduction**. Osteoarthritis (OA) is one of the most common causes of musculoskeletal system's ailments. In the prevention of the disease and in its comprehensive treatment, proper health-related behavior becomes an extremely important factor for maintaining an optimal health condition. The aim of the study is to assess the relationship between the reported pain and the disability level, and the health-related behaviors undertaken by OA patients as well as their acceptance of the disease.

**Materials/Methods**. The study group consisted of 198 patients with diagnosed OA, according to ACR criteria (1988). The method used in the study employed a Pain VAS (0-10), Health Assessment Questionnaire Disability Index (HAQ DI 0-3), Acceptance of Illness Scale (AIS 8-40) and Health and Behavior Inventory (IZZ 24-120).

**Results**. The average age among respondents with OA has been 59.16 years of age ($\pm$15.87), duration of disease 5.5 years ($\pm$4.32). Pain experienced both during movement ($r_s = 0.319$, $p < 0.001$) and at rest ($r_s = 0.382$, $p < 0.001$) correlated positively with physical disability (HAQ DI). Studies indicated a positive linear correlation between the age and physical disability ($r_s = 0.200$, $p = 0.005$). Acceptance of the disease (AIS) depends mostly on age ($r_s = -0.325$, $p < 0.001$), on pain in motion ($r_p = -0.209$, $p < 0.001$) and at rest ($r_p = -0.218$, $p < 0.001$) and on the disability levels ($r_p = -0.353$, $p < 0.001$). Analysis of the health-related behaviors (IZZ) indicates that the average severity of declared behavior is statistically significant with physical disability (HAQ DI) ($p = 0.029$).

**Conclusions**. The acceptance of illness is significantly reduced with age and progressive levels of disability as well as with the severity of pain. The progressive levels of disability and the younger age of the respondents motivate them to engage in health beneficial behavior.

## INTRODUCTION

Osteoarthritis—OA (*morbus degenerativus articulorum*), also known as a degenerative disease, is a group of overlapping disorders that, despite their different etiologies, lead to similar effects within the biological, morphological and clinical results. The disease affects ligaments, joint capsules, synovium, bursae, tendons, muscles, and it is often accompanied by the secondary damage to the nerves and veins (*Bannell, Hunter & Hinman, 2012*; *Jordan et al., 2003*). This applies in particular to the weight-bearing joints; for example, knees, ankles, spine and upper limbs, less often hips.

The main physical problems of OA patients are: pain, contracture and distortion of joints as well as difficulty with moving and during performing basic tasks of self-care. The pain caused by the disease contributes to the feeling of anxiety, irritability, exhaustion, which in turn interfere with the everyday life functioning (*Kool & Geenen, 2012*). The patients display behaviors resulting from the fear of losing physical mobility, such as depressive states, despondency, and reluctance to undertake any form of physical activity. Moreover, the progressive character of the disease causes problems within the social and professional spheres which lead to isolation, lack of acceptance of one's inability for professional work, restrictions in movement and limitations in performing basic daily living activities (*Østeraas et al., 2013*; *Kolanowski, 2010*; *Suri & Walsh, 2012*). Since for a large group of patients it is extremely difficult to adapt to the changes brought by the chronic, progressive disease it is important that they obtain professional help, support and health education. They should be prepared for a conscious participation in the treatment and self-care (*Hill, 2006*; *Sierakowska et al., 2010a*). Another key factor is laying the foundations for the proper health-related behaviors because that can reduce or even prevent the progression of disability and physical dysfunctions.

Health behaviors that positively affect health are primarily a healthy diet, regular physical activity and an optimum amount of sleep per day, as well as self-control of the health state, responsibility for one's own health and a positive attitude (*Banaszkiewicz & Andruszkiewicz, 2008*).

In osteoarthritis, particular attention is paid to health behaviors that improve the function of the musculoskeletal system. Patients have to take daily activity in the form of physical exercises tailored to their individual abilities, and daily walks to improve their overall condition. It is important to maintain a healthy diet in order to keep a low weight. Such actions help in reducing joint stress and pain (*Sierakowska et al., 2010a*). Given the adverse effect of non-steroidal anti-inflammatory drugs, it is recommended to use them mainly topically on the painful joint. Physiotherapy can also be helpful in relieving pain and disability (*Iwamoto et al., 2011*). The patient's active participation in the treatment together with maintaining healthy lifestyle and behaviors that reduce the disability progression will improve his/her everyday life functioning and, thus, the acceptance of disease will increase (*Jordan et al., 2003*; *Hochberg et al., 2012*; *Bannell, Hunter & Hinman, 2012*).

The authors aim to demonstrate the osteoarthritis not only from the biomedical perspective, but also as a psychological and existential problem of the patient who has been diagnosed with a chronic, progressive rheumatic disease. The applied research tools

are based on the social-cognitive theories and refer to the holistic approach to health and disease. Therefore, the evaluation of patient's behavior focuses on health, physical complaints, methods of coping with the disease and treatment and consider cognitive, emotional and motivational aspects. This is particularly important in osteoarthritis, as it can lead to the reduced level of performance, a high severity of health problems and dependence on the environment (*Juczyński, 1999*; *Felton, Revenson & Hinrichsen, 1984*; *Amir, 1987*; *Grohman, 1982*). The mode of dealing with the disease and initiating or discontinuing pro-health behaviors affects health related quality of life. In planning treatment, care and targeted health education it is very helpful to know how the patient is adapts to the disease and copes with it without experiencing negative emotions and rejection (*Juczyński, 1999*; *Hill, 2006*; *Sierakowska et al., 2010a*).

## THE AIMS OF THE STUDY

This study is an attempt to evaluate the degree of acceptance of the disease and pro-health behaviors, in relation to the major health problems in osteoarthritis patients, such as pain and inability to independently perform activities of daily living; to determine how selected demographic factors, such as age and sex, as well as the duration of disease, affect the acceptance of osteoarthritis and taking pro-health behaviors and, in consequence, the progression of pain and disability; an analysis of the actions undertaken by the patients that could improve their well-being and everyday functioning (patients taking analgesics, doing physical exercises on their own).

The researchers hypothesized that patients who engage in pro-health behaviors are able to better cope with pain and progressive physical disability. The health beneficial behaviors should improve the acceptance of the disease, which usually decreases with age and duration of osteoarthritis. An important task for health professionals is to motivate the patient to engage in behaviors that are health beneficial. The authors have attempted to identify the factors affecting the acceptance of disease and positive lifestyle, so that educational activities and support for OA patients could be planned deliberately and accurately.

## MATERIALS AND METHODS

### Participants and procedures

This cross-sectional correlational study included 198 patients diagnosed with osteoarthritis of the knee, osteoarthritis of the hip and degenerative disease of the spine, within the program of inpatient and outpatient care. The study was conducted at the Department of Rheumatology and Internal Diseases, Medical University of Bialystok, Poland, during patients' hospitalization, and at Rheumatology Outpatient Clinic in Augustow, during inspection visits to rheumatologists, from April to November 2011. The patients were informed about the study and instructed how to fill in the questionnaires independently and confidentially.

The inclusion criteria were: age $\geq 40$ years, diagnosis of OA according to ACR criteria (1988) and informed consent to participate in the study. The criterion for exclusion from

the study was the existence of other, overlapping diseases of bones and joints, including inflammatory joint diseases. Patients who met the criteria were qualified for the study according to the order of their admission to the Department of Rheumatology or visit in Rheumatology Outpatient Clinic.

## Ethics approval

The study follows the *Good Clinical Practice* guidelines and it is in accordance with the *1975 Helsinki Declaration* revised in 2000 (concerning the ethical principles for the medical community and forbidding the releasing of the patient name initials or the hospital evidence number) and with the ethical standards of the Institutional Committee on Human Experimentation (statute from the Bioethics Committee of the Medical University in Bialystok, Poland, no. RI-002/572/2011).

## Study instruments

The method used was a diagnostic survey using measurement tools employed in the promotion of health and health psychology that are accessible to health professionals, such as: the Acceptance of Illness Scale (AIS 8-40) by *Felton, Revenson & Hinrichsen (1984)* adapted by *Juczyński (1999)*, and the Health Behavior Inventory (IZZ 24-120) by *Juczyński (1999)*. Additionally, the Visual–Analog Scale Assessment of Pain (Pain VAS) was applied during movement and resting (0–10) as well as the Health Assessment Questionnaire Disability Index (HAQ DI 0-3).

The Acceptance of Illness Scale (AIS) has been used for studying patients. It contains eight statements that describe the negative consequences of ill health, taking into account the limitations imposed by the disease, lack of self-sufficiency, the sense of dependence on others and low self-esteem. The scale is used to measure the degree of disease acceptance (*Juczyński, 1999*). To evaluate the level of the acceptance of disease, the results were interpreted within the scale of 8–40 points. The higher the score, the greater acceptance of disease, the better adaptation to the limitations imposed by the disease and the lower the sense of psychological discomfort.

The Health Behavior Inventory (IZZ) is designed for studying healthy and ill adults. It comprises 24 statements that describe the intensity of health-related behaviors. The scale allows for the evaluation of the intensity of health-related behaviors in four areas (1-5): proper eating habits (type of food intake e.g., vegetables, fruit, whole wheat bread) preventive behavior (following doctor's recommendations, interest in knowledge about the disease), positive mental attitude (avoiding strong emotions and stress) and healthy practices (sleep, recreation, physical activity). IZZ is helpful in planning measures of prevention, behavior modification determining the direction and monitoring of changes in health practices (*Sierakowska et al., 2010a*). For the overall evaluation of health-related behaviors, the results are interpreted within the scale of 24–120 points. They can be converted into raw values standard ten (1–10), given the temporary standards for men and women (1–4 standard ten scores low F 24-77, M 24-71; average 5-6 F 78-91, M 72-86 7-10 92-120 high F, M 87 - 120) (*Juczyński, 1999*).

The severity of pain (Pain VAS 0-10) has been interpreted in three ranges: 0–3.5—a slight degree of pain (low); 3.6–6.5—an average pain (medium); 6.6–10—a strong degree of felt pain (strong) (*Wiland, Madaj & Szmyrka-Kaczmarek, 2008*).

The HAQ-DI evaluates the ability to perform daily activities during the last week. The questionnaire consists of 20 basic questions divided into eight sections, in which the patient has a choice of four possible answers: without difficulty, with certain difficulty, with difficulty, unable to perform, regarding the activities of everyday life functioning (dressing and washing, morning getting up, eating, walking, personal hygiene, lifting, gripping and movement). The questionnaire also includes additional questions regarding the assistive devices used to facilitate the functioning and the activities that require help of other people. The total score ranges from 0–3: 0–1—little degree dysfunctions in any field of daily life; >1–2—serious limitations or need for help in daily activities; >2–3—total inability to do daily activities without help (*Bruce & Fries, 2003*; *Thorsen et al., 2001*).

## DATA ANALYSIS

All data were analyzed using PQStat v.1.4.2 software. We tested the null hypothesis of no correlation between health behavior, acceptance of disease and patient pain problem and disability. Pearson ($r_p$) and Spearman ($r_s$) correlation coefficient is reported together with $p$-values, with $r$ of 0.10, 0.20 and 0.50 representing small, medium and large effects, respectively. The effects of sex, age and disease duration were tested across all measures. Students' $t$-test was used to assess gender differences and One-Way ANOVA for differences across age groups and disease duration. The level of significance $\alpha = 0.05$.

## RESULTS

### General characteristics of subjects with OA

Generally, as presented in Table 1, the largest group ($n = 110$) of patients diagnosed with OA was women (55.6%). The mean age was 59.16 ($\pm 15.87$). The average time of disease duration was 5.5 ($\pm 4.32$) years. More than half of respondents (56%) had OA longer than 10 years. As shown in Table 1, the largest group of patients ($n = 100$, 50.5%) declared primary education/vocational training and lived in the city ($n = 122$, 61.6%). The vast majority ($n = 138$, 70.1%) of subjects were retired and married ($n = 147$, 74.2%).

The majority of respondents ($n = 117$, 59%) were taking analgesics during the periods of the disease's worsening. The level of physical activity was not satisfactory. More than half of respondents ($n = 104$, 52.5%) declared that they did not practice any sport. A large percentage of respondents ($n = 88$, 44.7%) did not use any form of rehabilitation.

### The analysis of pain perception during motion and rest (Pain VAS)

The mean of pain during movement in the studied group of OA patients, as presented in Table 1, was 5.92 ($\pm 1.90$), and the rest 4.95 ($\pm 2.27$), which indicates the average level of pain. In detailed analysis of the data on the severity of pain during movement, it was shown that more than half of all patients ($n = 100$, 50.5%) declared a strong degree of experienced pain, 29.8% of patients ($n = 59$) declared pain while resting.

**Table 1  Patient characteristics and outcomes (mean (SD) except where stated otherwise).**

| Variables studied (score range) | Mean (±SD) |
|---|---|
| Age | 59.16 (±15.87) |
| Disease duration years | 5.5 (±4.32) |
| Sex—number of women (%) | 110 (55.6) |
| Educational background | |
|     Basic/ professional—number (%) | 100 (50.5) |
|     Secondary—number (%) | 61 (30.8) |
|     Higher—number (%) | 37 (18.7) |
| Place of residence | |
|     City—number (%) | 122 (61.6) |
|     Countryside—number (%) | 76 (38.4) |
| Occupational status | |
|     Retired—number (%) | 138 (70.1) |
|     Working—number (%) | 55 (27.9) |
|     Unemployed—number (%) | 5 (2.5) |
| Family status | |
|     Married/married—number (%) | 147 (74.2) |
|     Widow/widower—number (%) | 43 (21.7) |
|     Single—number (%) | 8 (4.0) |
| Pain-VAS (0-10) in motion | 5.92 (±1.90) |
| Pain-VAS (0-10) at rest | 4.95 (±2.27) |
| HAQ-DI (0-3) | 1.10 (±0.92) |
| AIS (8-40) | 25.75 (±8.47) |
| IZZ (24-120) | 88.39 (±15.5) |

**Notes.**

VAS, visual–analogue scale; HAQ DI, Health Assessment Questionnaire Disability Index; AIS, Acceptance of Illness Scale; IZZ, Health Behavior Inventory.

The statistical analysis showed a statistically significant relationship between the perception of pain during movement and taking analgesics. Patients who did not take analgesics rated their pain lower significantly more often—Pain VAS 3.94 (±1.81) ($p < 0.001$). Patients who declared average level of pain during movement—Pain VAS 6.10 (±1.91), more frequently admitted regular taking analgesics. As shown in Table 2, there is a statistically significant relationship between the level of pain at rest and analgesics intake ($p < 0.001$). With the increase of pain at rest the frequency of intake of analgesics has been intensified.

The answers on the character of pain in relation to the physical exercises at home (physiotherapy) suggest that the level of pain experienced both during movement and at rest was slightly reduced through performing physical exercises at home, although it was not statistically significant (data in Table 2).

**Table 2** The level of pain during movement and resting (Pain VAS) in comparison to the variables in the group with osteoarthritis.

| Variables studied | Pain in motion (VAS 0-10) | | Pain at rest (VAS 0-10) | |
|---|---|---|---|---|
| | Mean (±SD) | [a]F-statistic (p-value) | Mean (±SD) | [a]F-statistic (p-value) |
| **Sex** | | | | |
| F | 5.1 (±2.22) | 0.12 (0.694) | 5.0 (±2.35) | 0.12 (0.732) |
| M | 4.79 (±2.14) | | 4.88 (±2.17) | |
| **Age, years** | | | | |
| 40–60 | 4.62(±1.92) | | 4.83 (±2.17) | |
| 61–76 | 4.96 (±1.57) | 0.74 (0.708) | 5.08 (±2.36) | 0.22 (0.802) |
| ≥77 | 4.76 (±2.54) | | 4.94 (±2.38) | |
| **Disease duration, years** | | | | |
| 0–5 | 5.72 (±2.05) | | 4.37 (±2.30) | |
| 6–10 | 5.78 (±1.71) | 0.62 (0.539) | 4.86 (±2.03) | 2.37 (0.096) |
| >10 | 6.05 (±1.90) | | 5.22 (±2.31) | |
| **Intake of analgesics** | | | | |
| During worsening of symptoms | 6.10 (±1.75) | | 5.10 (±2.20) | |
| Systematically | 6.10 (±1.91) | 11.01 (**<0.001**) | 5.21 (±2.25) | 8.49 (**<0.001**) |
| Not taking | 3.94 (±1.81) | | 2.85 (±1.88) | |
| **Physical exercises** | | | | |
| Doesn't perform physical exercises | 5.87 (±2.08) | | 4.89 (±2.33) | |
| Several times a month | 6.42 (±1.60) | | 5.67 (±2.07) | |
| 2–3 times a week | 5.82 (±1.64) | 0.84 (0.471) | 4.76 (±1.99) | 1.19 (0.313) |
| Daily | 5.69 (±1.78) | | 4.66 (±2.58) | |

**Notes.**
[a]The univariate ANOVA for independent groups, F statistic.
VAS, visual—analog scale.

## The analysis of the level of physical disability in performing daily activities (HAQ DI)

In order to analyze the degree of patients' physical disability, the HAQ DI questionnaire was used. In the study group, as it is indicated in Table 1, the average HAQ DI score was at 1.10 (±0.92).

The average value level of disability among women was 1.25 (±1.07), while in men 0.92 (±0.64). The statistical analysis showed that there is a statistically significant relationship between the level of inability in performing daily activities and sex ($p = 0.012$) (data in Table 3).

The average level of disability in the age group ≥77 years was 1.22 (±0.72) (it was the highest value in all groups) ($p = 0.028$). The study has shown that more than half of the patients (62.4%) aged ≥77 years, declared major restrictions or the need for help in daily living activities (HAQ DI>1-2). The study indicated a positive linear correlation, showed in Table 3, between the age and physical disability ($r_s = 0.200$, $p = 0.005$).

The evaluation of skills in everyday life, as presented in Table 3, has been positive in patients who declared that they were not taking any analgesics (HAQ DI $0.59 \pm 0.43$). Respondents who regularly took analgesics obtained the highest level of disability (HAQ

**Table 3** The level of physical disability (HAQ DI) in comparison to the variables in the group with osteoarthritis.

| Variables studied | HAQ DI (0-3) | | |
|---|---|---|---|
| | Mean (±SD) | [a]F-statistic (*p*-value) | [b]$r_s$ (*p* − value) |
| **Sex** | | | |
| F | 1.25 (±1.07) | 6.38 (**0.012**) | |
| M | 0.92 (±0.64) | | |
| **Age, years** | | | |
| 40–60 | 0.98 (±1.17) | | |
| 61–76 | 1.18 (±0.64) | 1.37 (**0.028**) | 0.200 (**0.005**) |
| ≥77 | 1.22 (±0.72) | | |
| **Disease duration, years** | | | |
| 0–5 | 0.93 (±1.47) | | |
| 6–10 | 1.03 (±0.67) | 1.56 (0.211) | |
| >10 | 1.20 (±0.66) | | |
| **Intake of analgesics** | | | |
| During worsening of symptoms | 1.06 (±1.03) | | |
| Systematically | 1.31 (±0.73) | 4.49 (**0.012**) | |
| Not taking | 0.59 (±0.43) | | |
| **Physical exercises** | | | |
| Doesn't perform physical exercises | 1.17 (±1.11) | | |
| Several times a month | 1.16 (±0.60) | 0.80 (0.496) | |
| 2–3 times a week | 1.03 (±0.62) | | |
| Daily | 0.88 (±0.71) | | |
| **Pain VAS in motion** (0-10) | | | |
| Low | 0.81 (±1.90) | | |
| Medium | 1.01 (±0.64) | 18.50 (**<0.001**) | 0.319 (**<0.001**) |
| Strong | 1.25 (±0.68) | | |
| **Pain VAS at rest** (0-10) | | | |
| Low | 0.92 (±1.28) | | |
| Medium | 0.97 (±0.60) | 18.28 (**<0.001**) | 0.382 (**<0.001**) |
| Strong | 1.47 (±0.63) | | |

Notes.
[a]The univariate ANOVA for independent groups, [1] F-statistic.
[b]$r_s$ Spearman correlation.
HAQ DI, Health Assessment Questionnaire Disability Index.

DI $1.31 \pm 0.73$). There has been observed a statistically significant relationship between the level of disability in the performance of activities of daily life and the intake of analgesics ($p = 0.012$).

The study indicated a statistically significant correlation between the level of pain during movement and physical disability (HAQ DI) ($p < 0.001$), as presented in Table 3. Patients who declared strong level of pain also declared serious limitations on performing daily life activities (HAQ DI $1.25 \pm 0.68$). There was a positive linear correlation ($r_s = 0.319$, $p < 0.001$) between the Pain VAS and HAQ DI. The average value for the level of disability

among patients who declared a strong level of pain at rest was 1.47 ($\pm$0.63). It has been observed that along with mobility improvement, the level of pain decreased ($r_s = 0.382$, $p < 0.001$) (Table 3).

### Correlates of disease acceptance (AIS)

The average value level of the acceptance of disease in the study group, as presented in Table 1, was 25.75 ($\pm$8.47), which indicates the average level of acceptance of the disease among patients with diagnosed OA.

With age the level of acceptance of the disease significantly worsened. The results of statistical analysis showed that there was a statistically significant correlation between the level of acceptance of the disease and the age ($r_s = -0.325$, $p < 0.001$).

In the statistical analysis of the variable of disease duration and the level of acceptance of the disease, it was observed, as shown in Table 4, that along with the duration of OA the level of acceptance of the disease significantly decreases (>10 years—AIS 23.71 ($\pm$7.79)). The analysis indicated a statistically significant relationship between the variables ($p < 0.001$).

The patients who declared that they do not take any analgesics assessed the acceptance of the disease on a good level—AIS 30.64 ($\pm$9.30) and those who take analgesics systematically pointed to the average level of the disease acceptance—AIS 24.35 ($\pm$9.10) ($p = 0.023$) (data in Table 4).

As shown in Table 4., it has been observed a negative correlation ($r_p = -0.209$, $p < 0.001$) between the level of the disease acceptance and the degree of pain during movement. Along with the seriousness of pain the capacity to accept the disease decreased. A relation between the level of acceptance of disease and the degree of pain at rest ($r_p = -0.218$, $p < 0.001$) has been also demonstrated.

The results also indicate a negative linear correlation between the acceptance of illness and the level of disability (HAQ DI) ($r_p = -0.353$, $p < 0.001$). This proves that the higher the acceptance of OA, the lower the level of physical disability.

The average value for the level of acceptance of disease among those declaring a slight dysfunction in every area of everyday life (HAQ 0-1) was 28.75 ($\pm$8.53), and among patients reporting a total inability in performing activities of daily living (HAQ>2-3)—21.06 ($\pm$6.02) ($p < 0.001$) (data in Table 4).

### Correlates of the inventory of health-related behaviors (IZZ)

In the general analysis of inventory of health-related behaviors it has been observed an average intensity of declared behavior—IZZ 88.39 ($\pm$15.34) (Table 1).

As shown in Table 4, health behaviors in the group of women was 92.51 ($\pm$14.02), while in men it was 83.23 ($\pm$15.44) ($p < 0.001$). The detailed analysis showed that 61.8% of women and 42% of men reported a high occurrence of health-related behaviors.

Given the age factor, mean value of inventory of health behaviors in the group $\geq$77 years was the lowest, compared to other age groups, and was 84.43 ($\pm$15.34). The analysis showed a statistically significant relationship between the declared health behavior and the age of the patients ($p = 0.033$) (Table 4).

Analysis of health-related behaviors in relation to the applied physiotherapy at home, showed that patients performing physical exercises every day, declared a high intensity of

**Table 4  Correlates of disease acceptance and health behaviors in the group with osteoarthritis.**

| Variables studied | AIS (8-24) | | | IZZ (24-120) | |
|---|---|---|---|---|---|
| | Mean (±SD) | [a]F-statistic/$p$-value | [b]$r_s$/ [c]$r_p$ ($p$-value) | Mean (±SD) | [a]F-statistic/$p$-value |
| **Sex** | | | | | |
| F | 30.23 (±8.45) | 2.51 (0.115) | | 92.51 (±14.02) | 4.67 (**<0.001**) |
| M | 28.22 (±7.54) | | | 83.23 (±15.44) | |
| **Age, years** | | | | | |
| 40–60 | 28.47 (±7.84) | | | 87.25 (±16.85) | |
| 61–76 | 24.42 (±8.27) | 9.46 (**<0.001**) | [b]−0.325 (**<0.001**) | 91.84 (±12.75) | 3.47 (**0.033**) |
| ≥ 77 | 22.23 (±8.45) | | | 84.43 (±15.34) | |
| **Disease duration, years** | | | | | |
| 0–5 | 30.31 (±8.54) | | | 87.46 (±15.47) | |
| 6–10 | 26.05 (±8.25) | 11.11 (**<0.001**) | | 85.82 (±17.43) | 1.06 (0.349) |
| >10 | 23.71 (±7.79) | | | 89.46 (±14.46) | |
| **Intake of analgesics** | | | | | |
| During worsening of symptoms | 25.80 (±7.76) | | | 87.43 (±15.20) | |
| Systematically | 24.35 (±9.10) | 3.81 (**0.023**) | | 91.01 (±15.46) | 1.56 (0.213) |
| Not taking | 30.64 (±9.30) | | | 85.11 (±15.33) | |
| **Doing physical exercises** | | | | | |
| Doesn't perform physical exercises | 25.26 (±8.83) | | | 82.67 (±15.91) | |
| Several times a month | 26.92 (±8.22) | 0.38 (0.765) | | 90.85 (±10.67) | 13.31 (**<0.001**) |
| 2–3 times a week | 26.38 (±8.03) | | | 97.64 (±10.51) | |
| Daily | 25.48 (±8.17) | | | 94.51 (±14.02) | |
| **Pain VAS in motion (0-10)** | | | | | |
| Low | 29.82 (±8.70) | | | 87.83 (±15.50) | |
| Medium | 25.80 (±8.38) | 3.38 (0.036) | [c]−0.209 (<0.001) | 89.46 (±15.07) | 0.28 (0.753) |
| Strong | 24.68 (±8.23) | | | 87.52 (±15.96) | |
| **Pain VAS at rest (0-10)** | | | | | |
| Low | 27.18 (±8.66) | | | 86.60 (±15.26) | |
| Medium | 27.04 (±7.93) | 4.14 (**0.017**) | [c]−0.218 (**<0.001**) | 88.86 (±15.51) | 0.73 (0.482) |
| Strong | 22.55 (±8.10) | | | 89.81 (±15.30) | |
| **HAQ DI (0–3)** | | | | | |
| 0–1 | 28.75 (±8.53) | | | 87.81 (±15.26) | |
| >1–2 | 22.20 (±6.93) | 11.53 (**<0.001**) | [c]−0,353 (**<0.001**) | 87.12 (±15.33) | 3.59 (**0.029**) |
| >2–3 | 21.06 (±6.02) | | | 98.06 (±13.30) | |

**Notes.**
[a]The univariate ANOVA for independent groups.
[b]$r_s$ Spearman correlation.
[c]$r_p$ Pearson's correlation coefficient where 0.10, 0.20 and 0.50 represent small, medium and large effects respectively.
 HAQ DI, Health Assessment Questionnaire Disability Index; AIS, Acceptance of Illness Scale; IZZ, Health Behavior Inventory.

the declared pro-health behaviors—IZZ 94.51 (±14.02), while those who did not practice any sport pointed to medium/average occurrence of pro-health behaviors—IZZ 82.67 (±15.91) ($p < 0.001$) (data in Table 4).

In seeking the significant relationship between health behaviors (IZZ), and the level of disability (HAQ DI), presented in Table 4, we found that the patients declaring dysfunction

of slight intensity in every area of everyday life (HAQ DI 0-1) pointed to the average severity of health behaviors—87.81 ($\pm$15.26), while patients requiring total assistance in performing daily life activities (HAQ DI>2-3) declared a high intensity of health-related behaviors—98.06 ($\pm$13.30) ($p = 0.029$). The study showed no statistical significant linear correlation between health-related behaviors and the studied variables.

Separate calculation of the four categories of health behaviors (1-5), indicates that the average value for healthy eating habits was to 3.70 ($\pm$0.55), preventive behaviors—4.13 ($\pm$0.60), positive mental attitude—3.87 ($\pm$0.60), and health practices—3.76 ($\pm$0.60). The study has shown that the patients received the highest score in the category of preventive behaviors, regarding treatment compliance and obtaining information about health and disease, and the lowest in the category of proper eating habits (type of food they eat).

In the analysis of the relationship between the level of acceptance of the disease (AIS) and undertaking health-related behaviors (IZZ), there was no statistically significant dependence between two analyzed variables.

## DISCUSSION

Osteoarthritis is the most common rheumatic disease that leads to progressive disability, and it influences all spheres of the patient's life: the physical, psychological, social and occupational (*Bannell, Hunter & Hinman, 2012*; *Jordan et al., 2003*).

The progressive nature of osteoarthritis undoubtedly affects the level of disease acceptance and development of individual pro-health behaviors. It should be remembered that health beneficial activities promote better health, well-being and might affect further development of the disease and disability (*Sierakowska et al., 2010a*). An essential psychological factor that helps in coming to terms with the level of progressive disability and escalation of pain is the acceptance of disease.

Generally, in our study, it was observed that the acceptance of osteoarthritis was affected mainly by such factors as age, pain, disability, and disease duration. Taking pro-health behaviors depended greatly on the level of disability, age and sex. One of the manifestations of pro-health behaviors was performing physical exercises from two to three times a week. Considering the analysis of the main health problems, the study has shown a positive correlation between the perception of pain and the level of physical disability. The intensification of both variables impacted the intake of analgesics. The evaluation of the level of disability depended also on age and sex.

The dominant problem, from the patient's point of view, is pain experienced during performing physical activities and, to a lesser extent, during resting. The pain of the disease contributes to the feeling of anxiety, irritability, exhaustion, which in turn causes disturbances in the everyday life functioning (*Kool & Geenen, 2012*; *Chen et al., 2011*).

Study result indicate a negative linear correlation between the level of disease acceptance, the, pain felt during movement and at rest and the level of disability. Severe pain and progressive difficulty in daily functioning significantly influence the level of acceptance of the disease.

In terms of disability in OA patients *Cuperus et al. (2015)* showed that the progressive nature of the disease negatively impacts patients' functioning in everyday activities. As a

result, in most cases, they need the help of others on performing basic tasks, e.g., walking, eating, personal hygiene, or shopping. In our study it was observed that the main physical activities which require the help of other people are reaching, grasping, opening, receiving and handling things. It was observed that there was a statistically significant correlation between the perception of pain during movement and at rest, and the level of disability during the performance of daily life activities of (HAQ DI).

In the reviewed literature, there is a significant correlation between the level of pain and disability in OA patients. Pain created various limitations to varying degrees, not only in the performance of professional duties, but also in daily activities and in the pursuing of personal interests (*Jadhav et al., 2001*). *Reis et al. (2014)* indicate that in women diagnosed with OA, there is a significant relationship between the pain and the level of disability during performing basic daily life activities, which is similar to our study result.

Taking the age factor into account it can be noticed that the younger group had a higher degree of disease acceptance than the group of elderly people. It has been also observed that along with the disease duration the acceptance of health situation deteriorated and the patients presented worse adaptation and a greater sense of psychological discomfort. The study of *Creedon & Weathers (2011)* showed that patients with diagnosed OA are older and that they are able to more easily accept their health and adopt a positive attitude towards the disease. The researchers emphasize, however, that the relationship between the pain and the disease acceptance is a normal part of the aging process and it can significantly limit the patient's ability to independently perform activities of daily life. Nevertheless, *Baird (2003)* argues that women have greater difficulties in accepting their illness, disability and pain. In this study there are no significant differences concerning sex in analyzing variable acceptance of OA. It was also observed that there was a relationship between the disability during the performance of daily life activities, sex and age. Women rated their self-care ability worse (HAQ DI) than men. The review of the literature also pointed out the relationship between the level of physical ability and patients' sex (*Wilmańska & Gułaj, 2006*) and age. Studies by *Kool & Geenen (2012)* on OA patients, showed that >56% of patients older that, 77 years needed a constant regular care.

In our study, patients who experienced pain of a fairly large severity and who have difficulty in performing daily life activities more often take analgesics and non-steroidal non-inflammatory drugs. According to the recommendations for the therapeutic approach to OA on the basis of the American College of Rheumatology (ACR), European League Against Rheumatism (EULAR) and Osteoarthritis Research Society International (OARSI) recommendations, in case of treatment ineffectiveness, the recommended pharmacotherapy of pain is based on non-steroidal anti-inflammatory drugs such as paracetamol, at the lowest effective dose and for as short period of time as possible. The optimal therapeutic management of OA requires the combined use of non-pharmacological and pharmacological treatment. It should be noted that the literature review reveals that patients with osteoarthritis tend to overuse the aforementioned drugs (*Jordan et al., 2003*; *Hochberg et al., 2012*; *Zhang et al., 2008*). The patients with osteoarthritis overtake the non-steroidal anti-inflammatory drugs wanting to stimulate fast therapeutic effect, which only adds to the drugs' side effects. According to the authors, patients hold the false belief

about their positive effects on the course of the disease, not taking into account the adverse drug reactions (*Jordan et al., 2003*; *Zhang et al., 2008*).

In the prophylaxis and during treatment of the osteoarthritis it is also important to implement behaviors that are beneficial for health. Literature review shows that little physical activity and lack of motivation for regular exercising is a substantial problem in patients with OA., It is important to know what are the recommended types of physical exercises for the individual patient, how to effectively perform them, and how to combat the pain. It is advised that patients understand the benefits of physiotherapy, because many of them do not follow the physical treatment recommendations out of fear of exacerbating the pain (*Hill, 2006*; *Sierakowska et al., 2010a*; *Sierakowska et al., 2010b*). In our study, almost half of the patients did not use any form of rehabilitation.

There has been observed a statistically significant relationship between the declared health behavior and the age of patients. Patients aged 61–76 years compared with older and younger patients, pointed to the higher occurrence of pro-health behaviors. However according to the study of *Gignac et al. (2013)* middle-aged people are more satisfied with coping with the disease in comparison with the subjects who were healthy and older. According to the authors of this study, the results indicate that with age the physical ability deteriorates, which motivates patients to engage in pro-health behaviors. This is confirmed by the fact that patients who declared dysfunction of slight intensity in every area of everyday life pointed to medium/average occurrence of healthy behaviors, while patients who required total assistance in performing daily living activities declared a high occurrence of pro-health behaviors. The subjects with a higher level of pro-health behavior were involved in regular physical exercises that improved their physical ability and well-being. A study by *Hawker et al., (2011)* also showed the impact of pro-health behaviors on the progression of disability and everyday life functioning.

Detailed analysis of the categories of health behavior showed that the study group obtained the lowest average score in healthy eating habits, and the higher score in prevention. Nevertheless, according to Juczyński, women during menopause declared the overall behavior somewhat lower, especially for health practices, and the highest score, as in our study, was in prophylactics. Scores for pro-health behavior of adult men were lower than in women (*Juczyński, 1999*). Standards for osteoarthritis treatment emphasize the importance of self-care, proper lifestyle and rehabilitation. A patient who is able to self-manage his/her own life, accepts the disease and becomes independent, adapts to changing conditions and learns to live and work, despite the existing restrictions at home, at work and in the social environment (*Hill, 2006*; *Sierakowska et al., 2010a*; *Sierakowska et al., 2014*).

It is worth noting that the social situation of the elderly, i.e., the possibility of obtaining emotional support from the immediate environment, significantly affects the level of acceptance of the disease and disability. The task of the therapeutic team is not only administering effective treatment, but also providing support and advice on how to handle problems of everyday life, as well as stress and limitations caused by the disease (*Long et al., 2002*; *Tak & Laffrey, 2003*; *Sierakowska et al., 2010b*).

### Study limitations

The study of patients with osteoarthritis has its limitations due to the applied research tool questionnaire that is based on the patients' assessment of their own health; thus, independent verification of data is impossible. The correlational design prevents causal interference. Another limitation of the study stems from the fact that it was been conducted in a particular part of Poland (Podlasie voivodeship); therefore, the results do not refer to the entire population of Polish patients diagnosed with OA.

## CONCLUSIONS

1. Although women declare slightly higher difficulties in everyday activities than men, they exhibit more positive health conducive behavior.
2. With age and progressive levels of disability as well as with the severity of pain, the acceptance of disease is significantly reduced.
3. The progressive levels of disability and the younger age of the patients motivate them to engage in health beneficial behaviors. The subjects present positive pro-health behavior and undertake regular physical exercises.
4. A high intensity of pain and a progressive disability impact patients' decisions to follow treatment recommendations regarding analgesics.

The study has shown the need for taking measures aimed at stimulating patients' motivation to improve their physical ability and health education. In particular, elderly people should be more engaged in daily physical activity. The challenge for health professionals is to fight off pain caused by arthritis, primarily through the use of non-pharmacological methods of pain management, as well as with a greater access to the various forms of rehabilitation. In planning the health education, the attention should be paid also to the pro-health dietary advice.

Further work is planned to develop and implement an education program to promote healthy, active lifestyles and rehabilitation for patients with osteoarthritis, especially for patients with long disease duration and the elderly, with the level of disability HAQ DI >1 and Pain VAS >5 cm. It will be also important to motivate men to participate in organized educational activities as well as in rehabilitation. Three and six months after the program ending, we plan to evaluate the effectiveness of pro-health behaviors undertaken independently by the patients and their impact on the evaluation of pain, disability and the disease acceptance while using standardized measurement tools.

### Funding

The authors received no funding for this work.

### Competing Interests

The authors declare there are no competing interests.

## Author Contributions

- Matylda Sierakowska and Izabela Wysocka-Skurska conceived and designed the experiments, performed the experiments, analyzed the data, contributed reagents/materials/analysis tools, wrote the paper, prepared figures and/or tables, reviewed drafts of the paper.
- Wojciech Kułak conceived and designed the experiments, performed the experiments, analyzed the data, contributed reagents/materials/analysis tools, prepared figures and/or tables, reviewed drafts of the paper.

## Data Availability

The raw data has been supplied as a Supplementary File.

## Supplemental Information

Supplemental information for this article can be found online at http://dx.doi.org/10.7717/peerj.3276#supplemental-information.

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
