# Peer review of "Identification of demographic factors and health problems that affect the acceptance of disease and health behaviors of patients with osteoarthritis"

_PeerJ, doi:10.7717/peerj.3276_

## Round 0.1 · original submission · Major Revisions

· Academic Editor

Major Revisions

Thank you for your article on identification of demographic factors and health problems that affect the acceptance of disease and health behaviors of patients with osteoarthritis. Your manuscript would benefit from the assistance of an English language editor who could help with grammatical errors and sentence structure. It appears that you have tried to include a lot of information in one manuscript and may benefit from focusing on data for your three specific aims. It would improve your manuscript to discuss implications of your research for clinical practice as well as research. Please address the concerns of both reviewer one and two.

Reviewer 1 ·

Basic reporting

Overall, this manuscript covers and interesting topic but lacks clarity and focus. The introduction does not appropriately set up the aims of the study. The results presented are excessive (9 pages of text excluding tables) and appear as every possible analysis was conducted. The authors should clarify the main 1-3 aims of the paper and focus on those analyses and results. Clarifying the purpose would also help to narrow the discussion of the results in the discussion section.
Introduction:
- Unclear what they authors mean on line 49 with “the concept of health-related behaviors is not clear.”
- Self-efficacy is one’s confidence in their ability to perform or engage in a behavior not “being ready to gradually overcome the more difficult tasks or taking new types of action.” Additionally, self-efficacy has absolutely nothing to do with the current aims of the study and is not being assessed – thus, should not be discussed.
- Aims – why were those specific demographic and health-related factors chosen?

Experimental design

Aims and purpose of the study need to be clarified

Methods:
- How were participants recruited? Did anyone approached say no to participation?
- How and when were participants asked to complete the assessments?
- Lines 128 – what groups are being analyzed and why those groups? Why not combine all participants, particularly if there are no differences between groups.

Validity of the findings

Discussion:
- The discussion includes several pages of material that are not related to the discussion of the results of the trial. When the results are discussed, they are presented in a fragmented way rather than adequately pulling together a comprehensive overall discussion highlighting the main takeaways and relating them to previous literature. I believe this is a consequence of having so many results presented.

Reviewer 2 ·

Basic reporting

The paper could benefit from an English language editor as the writing is cumbersome and verbose, some sentences are unclear, and there are grammatical errors. Some terms should be changed, such as “diagnosed with” instead of “suffer from”, “analgesics” instead of “pain killers”. Some terminology is inconsistent making the paper difficult to follow, e.g., physical function vs. functional capacity vs. physical disability vs. physical efficiency and acceptance vs. tolerance.

An Introduction is provided with an overview of osteoarthritis and the importance of understanding the outcomes of pain, physical function, and health behaviors of patients with osteoarthritis. The authors should consider moving the beginning part of the Discussion to the Introduction as it would fit better in the Introduction. The authors mention self-efficacy in the Introduction section and social cognitive theory in the Discussion section, and they cite Bandura in both sections. However, they do not measure self-efficacy. Readers may benefit from inclusion of a study model showing the demographic variables (sex, age), patient characteristics (disease duration, intake of analgesics, physical exercise), clinical outcomes (pain [in motion, at rest], physical function), beliefs (disease acceptance), and health behaviors and the proposed relationships among the variables. The paper is well referenced with both classic and current references and includes published clinical guidelines.

A standard structure is used with Abstract (structured format), Introduction, Aims, Materials and Methods, Data Analysis, Results (with subsections), Discussion, Conclusions, and References. Four tables are embedded in the narrative when they are first mentioned. The tables are clear with footnotes. Missing data on Occupational Status (5 cases) in Table 1 are not reported and should be added. Raw data were provided, but this reviewer could not open the file.

The paper appears to report the full results of the study, and the results are relevant to the stated aims, although additional results are presented as well.

Experimental design

The study fits within the health sciences aims and scope of PeerJ. The authors do not explicitly state a research design; however, the design is a cross-sectional correlational design. Three aims are stated. However, results are presented that do not align with the aims as the authors report correlates of pain and disability in addition to correlates of disease acceptance and health behaviors. The authors identify the gap in knowledge as factors affecting disease acceptance and health behaviors in the population of adults with osteoarthritis to inform future educational interventions.

The study appears to have been well conducted. It would help readers if there were subsections in the Materials and Methods section, such as a Sample section, Measures section, and Procedures section. Additional information about the reliability and validity of the measures should be provided. The procedures need to be better explained for the study to be replicated. The study had Human Subjects approval.

Validity of the findings

A concern with the analyses is that multiple testing was done. Multiple linear regression analysis models on pain, physical function, disease acceptance, and health behavior could be done instead of multiple univariate ANOVA models and Spearman’s rho correlations. If linear regression assumptions cannot be met, then logistic regression models could be run. Some of the correlates could be entered into the regression models as continuous level variables instead of categorical variables, e.g., age and disease duration. It is not clear what data are “qualitative” (page 6, line 125). The authors report doing Chi Square analyses but it is not clear in the table footnotes that these analyses were done. Raw data were provided as noted above.

It would help readers if the subsections in the Results section were titled by concepts instead of by measures. For example, “Correlates of Disease Acceptance” could be used instead of “Analysis of Acceptance of Illness Scale (AIS)”.

The Discussion section should be revised to begin with a summary of the key results and should be reorganized to align with the aims. The authors compare the results to previous findings. Additional study limitations should be added, such as the correlational design preventing causal inference. Some speculation about the meaning of the results is found in the Conclusions section.

Additional comments

It would improve the paper if the authors expanded on the implications of the findings for research and clinical practice. Do the findings support the social cognitive theory that was used? How can the findings be used to design educational programs for patients with osteoarthritis? What are the next steps for future research?

---

## Round 0.2 · Minor Revisions

· Academic Editor

Minor Revisions

The manuscript is much improved, but still needs work in regards to the English language and structure. Another review by an English language editor is important.

In addition to the comments from both reviewers, I have some additional observations:

Line 27, change bursitis to bursae.
Line 239, as this study does not show cause, it is incorrect to state that it has also been proven. I would use verbiage such as shown.
Lines 255-257, I do not see the data that is listed here in Table 3, but rather Table 4.

Please address the comments of the 2 reviewers.

Reviewer 1 ·

Basic reporting

- Several grammar issues are present throughout the paper (i.e., line 66, “It has been used a study of healthy psychology…”; Lines 189-190 “In order to analyze the degree of physical disability of patients in everyday activities, there has been used a HAQDI questionnaire.” These issues make it challenging to read.

- Citations should be added in lines 45-51.

- Additional justification is still needed to clarify why acceptance of the illness is important and why the authors believe that will be related to health behaviors.
- I believe the introduction abruptly discusses psychological constructs, but there is no transition or justification for why. I think te paragraph in lines 64-72 could be shortened and combined with the introduction before the aims of the study.
- Lines 75-77 – unclear if this is really a hypothesis as well as what part of the aim of the study this connects to

Experimental design

- Although the authors attempt to improve the clarity of the paper, the revised manuscript is difficult to follow. The introduction needs to better justify the aims and then the results need to directly and clearly match up to the aims.

- How the 198 patients were recruited is still unclear, particularly as the response to the initial question about recruitment just repeated exactly what was already in the paper. How were those 198 chosen and not others?

Validity of the findings

- Lines 165-166 – It is unclear who the “test” and “rest” groups are.

Additional comments

- It is unclear on line 46 what “Patients have to take daily physical activity…” means. Do the authors mean patients need to engage in daily physical activity?

- The first two paragraphs in the discussion are not necessary, or could be integrated within the introduction.

Reviewer 2 ·

Basic reporting

The English language in the revised paper is improved over the first submission, but another review by an English language editor is needed to correct sentences that are awkwardly written and grammatically incorrect. In addition, make the following editorial changes:
• On line 65, change “suffers from” to “diagnosed with.” On line 150, change “suffered more” to “had OA longer than 10 years.”
• Replace “researches” with “studies” in the abstract and line 198.
• Change “anti-pain drugs” to “analgesics” on lines171, 174, 176, and 201.
• For consistency, replace “tolerance” with “acceptance” in the abstract in two places.
• On line 27, replace “bursitis” with “bursa.”
• On line 49 replace “taking the relief of the joints and reducing pain” to “reducing joint stress and pain.”
• On line 50, the term “locally” is unclear so please revise.
• On line 91, change “anonymously” to “confidentially.”
• On line 103, change “instrument” to “instruments.”
• On lines 104 to 109, change the order of the instruments to match the description provided in the paragraphs that follow: AIS, IZZ, Pain VAS, HAQ-DI.
• On line 114 and 125, change “pts” to “points.”
• On lines 126-127, “sten” is unclear; the information runs together and is not in consistent order making it difficult to follow, so pleaser revise.
• On lines 156, 157, and 168, report the n along with the percentage.
• On line 159, delete “numbers.”
• On line 179, change “the” to “at” so it reads “ and at rest.”
• On lines 242 and 243, “pkt” is unclear.
• On line 255, change “Table 3” to “Table 4.”
• On line 295, replace the second “such” with “as” so it reads “such factors as…”
• On line 310, “test” is unclear.
• On line 313, delete “that,” so it reads “observed that the.”
• On line 337, delete “to” so it reads “pointed out the.”
• On line 376, revise the sentence so it read “A study by Hawker et al. (2011) also showed...”
• On line 384, change “lives” to “life.”

An Introduction is provided with an overview of osteoarthritis and the importance of understanding the factors that affect acceptance of the disease and health behaviors by patients with osteoarthritis. The authors mention social cognitive theories in the Introduction section as the theoretical basis for the study; however no citation is given. The authors indicate the variables being examined in the study: demographic variables (sex, age), patient characteristics (disease duration, intake of analgesics, physical exercise), clinical outcomes (pain [in motion, at rest], physical function), beliefs (disease acceptance), and health behaviors and the proposed relationships among the variables. The paper is well referenced with both classic and current references and includes published clinical guidelines.

A standard structure is used with Abstract (structured format), Introduction, Aims, Materials and Methods (with subsections), Data Analysis, Results (with subsections), Discussion, Conclusions, and References. Four tables are embedded in the narrative when they are first mentioned. The tables are clear with footnotes and consistent with the narrative report of the findings. Raw data were provided, but this reviewer could not open the file. The paper appears to report the full results of the study, and the results are relevant to the stated aims; ancillary findings were deleted in the revised paper.

Experimental design

The study fits within the health sciences aims and scope of PeerJ. The authors do not explicitly state a research design, which should be added on line 85: “This cross-sectional correlational study included 198 patients…” Three aims are stated on lines 56-63. The authors identify the gap in knowledge as factors affecting disease acceptance and health behaviors in the population of adults with osteoarthritis to inform future educational interventions. The study appears to have been well conducted. The procedures were expanded in the revised paper. Additional information about the reliability and validity of the AIS, IZZ, and HAQ-DI should be provided.

Validity of the findings

The description of the data analyses was revised and is appropriate for the aims. Raw data were provided as noted above. The Discussion section was revised to summarize the key results and was reorganized to align with the aims. The authors compared their results to previous findings. The conclusions are well stated and do not go beyond the results. An additional study limitation should be added that the correlational design prevents causal inference.

Additional comments

The authors expanded on the implications of the findings for research and clinical practice.

---

## Round 0.3 · accepted · Accept

· Academic Editor

Accept

Thank you for your updated version of your manuscript. It has greatly improved.